# Exercise promotes the expression of brain derived neurotrophic factor (BDNF) through the action of the ketone body β-hydroxybutyrate

Sama F Sleiman[1]*, Jeffrey Henry[2,3,4,5], Rami Al-Haddad[1], Lauretta El Hayek[1], Edwina Abou Haidar[1], Thomas Stringer[2,3,4,5], Devyani Ulja[2,3,4,5], Saravanan S Karuppagounder[6,7], Edward B Holson[8,9], Rajiv R Ratan[6,7], Ipe Ninan[2,3,4,5], Moses V Chao[2,3,4,5]*

[1]Department of Natural Sciences, Lebanese American University, Byblos, Lebanon; [2]Skirball Institute of Biomolecular Medicine, New York University Langone Medical Center, New York, United States; [3]Department of Cell Biology, New York University Langone Medical Center, New York, United States; [4]Department of Neuroscience and Physiology, New York University Langone Medical Center, New York, United States; [5]Department of Psychiatry, New York University Langone Medical Center, New York, United States; [6]Burke Medical Research Institute, White Plains, United States; [7]Brain Mind Research Institue, Weill Medical College of Cornell University, New York, United States; [8]Stanley Center for Psychiatric Research, The Broad Institute of MIT and Harvard, Cambridge, United States; [9]Atlas Venture, Cambridge, United States

*For correspondence: sama. sleiman01@lau.edu.lb (SFS); moses.chao@med.nyu.edu (MVC)

**Abstract** Exercise induces beneficial responses in the brain, which is accompanied by an increase in BDNF, a trophic factor associated with cognitive improvement and the alleviation of depression and anxiety. However, the exact mechanisms whereby physical exercise produces an induction in brain *Bdnf* gene expression are not well understood. While pharmacological doses of HDAC inhibitors exert positive effects on *Bdnf* gene transcription, the inhibitors represent small molecules that do not occur *in vivo*. Here, we report that an endogenous molecule released after exercise is capable of inducing key promoters of the *Mus musculus Bdnf* gene. The metabolite β-hydroxybutyrate, which increases after prolonged exercise, induces the activities of *Bdnf* promoters, particularly promoter I, which is activity-dependent. We have discovered that the action of β-hydroxybutyrate is specifically upon HDAC2 and HDAC3, which act upon selective *Bdnf* promoters. Moreover, the effects upon hippocampal *Bdnf* expression were observed after direct ventricular application of β-hydroxybutyrate. Electrophysiological measurements indicate that β-hydroxybutyrate causes an increase in neurotransmitter release, which is dependent upon the TrkB receptor. These results reveal an endogenous mechanism to explain how physical exercise leads to the induction of BDNF.

## Introduction

It has been traditionally thought that physical activity and the processes of learning and memory formation are independent and carried out by different organ systems. However, from an evolutionary perspective, these processes needed to be tightly intertwined to ensure the survival of animal

**eLife digest** Exercise is not only good for our physical health but it benefits our mental health and abilities too. Physical exercise can affect how much of certain proteins are made in the brain. In particular, the levels of a protein called brain derived neurotrophic factor (or BDNF for short) increase after exercise. BDNF has already been shown to enhance mental abilities at the same time as acting against anxiety and depression in mice, and might act in similar way in humans. Nevertheless, it is currently not clear how exercise increases the production of BDNF by cells in the brain.

Sleiman et al. have now investigated this question by comparing mice that were allowed to use a running wheel for 30 days with control mice that did not exercise. The comparison showed that the exercising mice had higher levels of BDNF in their brains than the control mice, which confirms the results of previous studies. Next, biochemical experiments showed that this change occurred when enzymes known as histone deacetylases stopped inhibiting the production of BDNF. Therefore Sleiman et al. hypothesised that exercise might produce a chemical that itself inhibits the histone deacetylases.

Indeed, the exercising mice produced more of a molecule called β-hydroxybutyrate in their livers, which travels through the blood into the brain where it could inhibit histone deacetylases. Further experiments showed that injecting β-hydroxybutyrate directly into the brains of mice led to increase in BDNF.

These new findings reveal with molecular detail one way in which exercise can affect the expression of proteins in the brain. This new understanding may provide ideas for new therapies to treat psychiatric diseases, such as depression, and neurodegenerative disorders, such as Alzheimer's disease.

species. Indeed, physical effort usually occurred in response to an imminent danger. Responding to that danger not only required running, but also necessitated better functioning of the brain through increased plasticity in order to adapt to new sources of stress, to learn to avoid dangers or better respond to them, and to map surroundings and learn the locations of hazards (*Noakes and Spedding, 2012*) All of these responses require improved memory.

Physical exercise produces many benefits in the brain that enhance cognitive function, blood flow and resistance to injury. One mechanism to account for the changes in brain plasticity is through the action of growth factors (*Cotman et al 2007*). A major contributor to the processes of learning and memory formation involves brain derived neurotrophic factor (BDNF) signaling pathways. It has been known for over two decades that physical activity or neuronal activity markedly enhances *Bdnf* gene expression in the brain (*Isackson et al., 1991*; *Neeper et al., 1995*) and that this increase in BDNF protein leads to activation of signaling pathways that result in exercise-dependent enhanced learning and memory formation (*Vaynman et al., 2004*). Though these results are widely recognized, it is important to note that very little is known about the molecular mechanisms that link exercise and *Bdnf* expression. Regulation of *Bdnf* expression occurs by many means, but how exercise influences the expression of trophic factors is not understood. In this paper, we are interested in understanding how physical exercise induces *Bdnf* gene expression. This is a significant question, since cognitive ability and synaptic plasticity are influenced by the levels of BDNF (*Lu et al., 2013*; *Park and Poo, 2013*; *Vaynman et al., 2004*) and BDNF signaling is reduced in many neurodegenerative and psychiatric diseases (*Autry and Monteggia, 2012*; *Zuccato and Cattaneo, 2009*).

During development, BDNF is required for the survival of specific neuronal populations and it participates in axonal and dendritic growth and synaptogenesis (*Alsina et al., 2001*; *Bibel and Barde, 2000*). A number of studies have indicated that decreased levels of BDNF are associated with depression and become enhanced following antidepressant treatment (*Duman and Monteggia, 2006*; *Martinowich et al., 2007*). Moreover, exercise frequently leads to an increase in BDNF in the central nervous system to promote improvement in cognitive ability and depressive-like behavior (*Marais et al., 2009*; *Russo-Neustadt et al., 2000*). Indeed, physical activity has been shown to have anti-depressant effects and to improve outcomes in animal models and for patients with

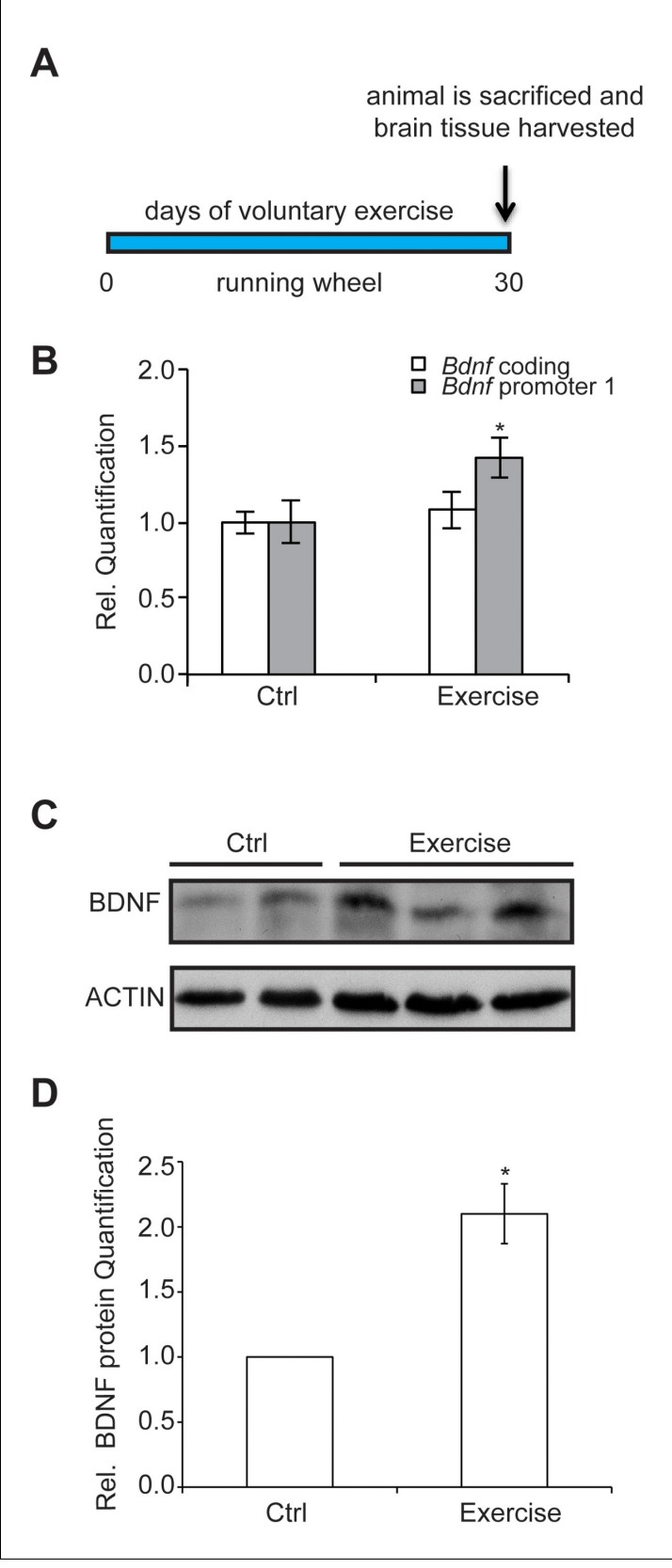

**Figure 1.** The experimental design and time course of the exercise regime is shown. Exercise induces changes in brain BDNF levels in a voluntary exercise protocol (16). (**A**) Experimental design for the Voluntary exercise model.
*Figure 1 continued on next page*

*Figure 1 continued*

(**B**) Voluntary exercise for 4 weeks significantly induces *Bdnf* promoter I expression in the hippocampus as measured by real-time RTPCR. The number of animal used for each group (control and exercise) is 10. *p<0.05 as measured by unpaired t-test. (**C**) Western blot analysis depicting the increase in mature BDNF protein levels in the hippocampus of exercise animals as compare to wild type. In this representative image, the BDNF levels from 2 control hippocampal lysates and 3 exercise hippocampal lysates are depicted. This experiment was replicated from additional 3 different animals in each group. (**D**) Quantification of the BDNF western blot.

neurodegenerative diseases such as Parkinson's Disease (*Frazzitta et al., 2014*) or Alzheimer's disease (*Smith et al., 2014*). As a result, by understanding the molecular mechanisms by which exercise induces *Bdnf* expression, we aim to harness the therapeutic potential of physical exercise and eventually identify novel therapeutic targets for both psychiatric and neurodegenerative diseases.

In animal models, exercise induces *Bdnf* mRNA expression in multiple brain regions (*Cotman et al., 2007*), most prominently in the hippocampus. BDNF production provides trophic support and increases in synaptogenesis and dendritic and axonal branching and spine turnover. Blocking BDNF signaling attenuates the exercise-induced improvement of spatial learning tasks (*Vaynman et al., 2004*), as well as the exercise-induced expression of synaptic proteins (*Vaynman et al., 2006*). However, how BDNF is selectively increased after physical activity-dependent changes in the nervous system is not well understood.

One mechanism that has been proposed is that exercise induces *Bdnf* expression through the induction of expression of Fndc5 (*Wrann et al., 2013*), a PGC-1α-dependent myokine. This hypothesis proposes that the FNDC5 protein is cleaved into a small circulating protein called irisin, which has been associated with the browning of fat (*Bostrom et al., 2012*). However, there are contradictory reports about whether *Fndc5* is translated and expressed at high levels after exercise and whether irisin is produced and found in blood (*Albrecht et al., 2015*; *Jedrychowski et al., 2015*). This raises questions about how and whether a myokine can be responsible for the induction of *Bdnf* gene regulation.

A second hypothesis is that exercise may induce BDNF levels by altering the epigenetic landmarks of the *Bdnf* promoters (*Guan et al., 2009*; *Koppel and Timmusk, 2013*). Because exercise induces metabolic changes and because epigenetics lies at the interfaces between the environment and changes in gene expression, it is conceivable that an endogenous molecule is produced after exercise, which can serve as a metabolite as well as a regulator of *Bdnf* transcription. In this paper, we provide a mechanism demonstrating that exercise induces the accumulation of a ketone body (D-β-hydroxybutyrate or DBHB) in the hippocampus, where it serves both as an energy source and an inhibitor of class I histone deacetylases (HDACs) to specifically induce BDNF expression.

## Results

### Exercise induces Bdnf expression in the hippocampus

To assess how exercise enhances *Bdnf* gene expression, we established a voluntary running protocol for mice (4 weeks of age), which has been previously shown to mediate increases in BDNF (*Marlatt et al., 2012*). Mice were individually housed in cages and divided into two groups: control and exercise. Each mouse in the exercise group was provided with a running wheel, whereas the control group mice were not. Mice were not forced to run on the wheels, but rather were allowed to run voluntarily. Long term running for up to 30 days was monitored (*Figure 1A*) for distance and time. We chose this exercise model because of its voluntary aspect. Stress has been reported to decrease BDNF levels (*Murakami et al., 2005*). For that reason, we avoided any handling or stress induction in the mice. Animals were sacrificed and tissues were collected for analysis. To determine the effects of voluntary running in the hippocampus, we initially focused our study on selective promoters from the *Bdnf* gene, which has a complex structure containing multiple promoters that generate many transcripts with a common coding exon (*Pruunsild et al., 2011*). Promoter I (pI), a neuronal activity dependent promoter (*Tabuchi et al., 2002*) and other nearby promoters (pIIA, pIIB, pIV) (*Pruunsild et al., 2011*), were also assessed for mRNA expression. Our results showed that voluntary exercise for 4 weeks significantly induced BDNF promoter I and II expression in the

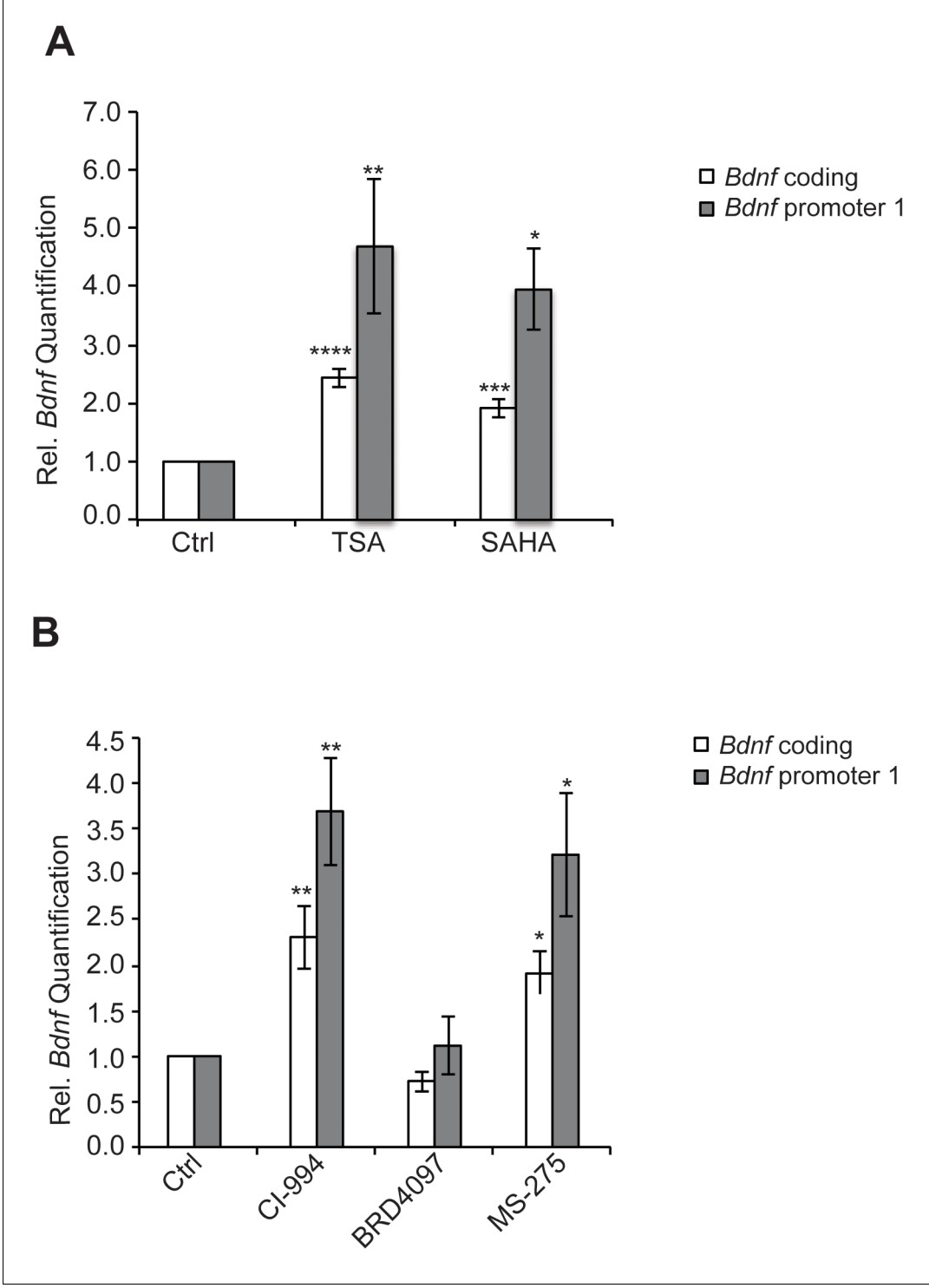

**Figure 2.** HDAC inhibitors induce *Bdnf* expression. (**A**) Broad spectrum HDAC inhibitors such as TSA (0.67 μM) and SAHA (5 μM) induce coding and pI *Bdnf* expression levels as measured by real-time RTPCR. For the coding promoter, the n number for controls, TSA and SAHA treatments are 6, 5 and 4 respectively. For the pI promoter, the n number for controls, TSA and SAHA treatments are 5, 5 and 4 respectively. Each replicate consisted of primary neurons obtained from different set cultures and treated with fresh dilutions of the compounds. Significance was measured by 1way anova **p< 0.01.and ***p<0.001. (**B**) Treatment with class I HDAC inhibitors such as CI-994 (10 μM) and MS-275 (10 μM) induce coding and pI *Bdnf* levels, whereas the negative control analog for CI-994, BRD4097 does not as measured by real-time RTPCR. For the coding promoter, the n number for

*Figure 2 continued on next page*

*Figure 2 continued*

controls, CI-994, BRD4097 and MS-275 treatments are 6, 6, 5 and 5 respectively. For the pI promoter, the n number for controls, CI-994, BRD4097 and MS-275 are 5, 6, 5 and 5 respectively. Each replicate consisted of primary neurons obtained from different cultures and treated with fresh dilutions of the compounds. Significance was measured by 1way anova **p<0.01.and ***p<0.001.

hippocampus (*Figure 1B* and data not shown). Western blot analysis of the hippocampus of animals after voluntary running showed a significant increase of mature BDNF protein levels, as compared to control mice (*Figure 1C and D*).

## Exercise induces changes in class I HDAC expression and binding to the hippocampal Bdnf promoter

One epigenetic mechanism that has been proposed for inducing *Bdnf* gene expression is histone acetylation (*Koppel and Timmusk, 2013*). A number of studies have reported that blocking HDACs enhances memory formation and the expression of synaptic plasticity genes, such as *Creb, Bdnf* and *CamkII* (*Guan et al., 2009*; *Koppel and Timmusk, 2013*). HDACs catalyze the deacetylation of histones and are divided into four classes. Class I (HDAC1, HDAC2, HDAC3 and HDAC8), II and IV are zinc dependent enzymes and are represented by at least 11 different proteins, whereas class III includes the sirtuins (*Haberland et al., 2009*).

We have found that the general class I/IIb HDAC inhibitor, SAHA (vorinostat), a clinically approved anticancer agent and memory enhancer (*Guan et al., 2009*) and TSA (trichostatin A), a broad-spectrum inhibitor, were effective in inducing *Bdnf* mRNA in primary cortical neuron cultures (*Figure 2A*). More specific HDAC inhibitors, such as CI-994 and MS-275, that target class I HDAC members were also capable of elevating *Bdnf* transcripts (*Figure 2B*), whereas a non-active analog of CI-994, BRD4097 (manuscript submitted), did not give a response. While these results implicate class I HDAC inhibition in the regulation of synaptic plasticity genes, such as *Bdnf*, it is not clear how exercise is translated into increases in BDNF. Interestingly, we found that exercise reduces *Hdac2* and *Hdac3* mRNA levels, but not *Hdac1* levels in the hippocampus (*Figure 3A*). This is consistent with the observation that HDAC2, but not HDAC1, is more important in binding to promoters of activity-related genes (*Guan et al., 2009*). These results suggest that exercise may modulate *Bdnf* gene expression in the hippocampus by inducing changes to the epigenetic landscape of its promoter. To test this possibility, we performed chromatin immunoprecipitation experiments and showed that the binding of both HDAC2 and HDAC3 to the *Bdnf* promoter in hippocampi of exercise animals was decreased, in comparison to sedentary animals (*Figure 3B* and data not shown). Even though some variation was detected in the basal binding of HDAC3 to the *Bdnf* pI promoter, the overall data is consistent with exercise inducing a significant decrease in HDAC3 binding to the *Bdnf* promoter. As a result, these results focused our search for endogenous metabolites that are produced in response to physical activity and that may also act as an epigenetic modulator.

## Exercise induces DBHB levels in the hippocampus

Exercise is accompanied with increases in energy requirements leading to mitochondrial biogenesis, as well as higher metabolism and oxygen consumption. For example, exercise is often associated with increases in ketone body production. The ketone body D-β-hydroxybutyrate (DBHB) is a major energy metabolite that is increased in the liver after prolonged exercise. DBHB levels are frequently increased after caloric restriction, fasting and ketogenic diets and DBHB is believed to serve as a signaling molecule in response to metabolic changes (*Newman and Verdin, 2014*). DBHB is synthesized from acetyl-CoA generated from the β-oxidation of fatty acids in the liver. As DBHB production is sensitive to environmental factors, nutritional states and energy levels, it has been hypothesized that ketone bodies may serve as an intermediary to regulate gene expression and chromatin structure. Interestingly, DBHB is transported in the blood stream to the brain where it can serve as an energy source. In addition, DBHB also has been shown to act as a class I HDAC inhibitor in non-neuronal tissues (*Shimazu et al., 2013*). To assess the effects of voluntary running, we measured DBHB levels in the hippocampus following exercise. We found that an increase in DBHB levels occurred in mice after exercise, compared to mice not subjected to voluntary running (*Figure 4A*).

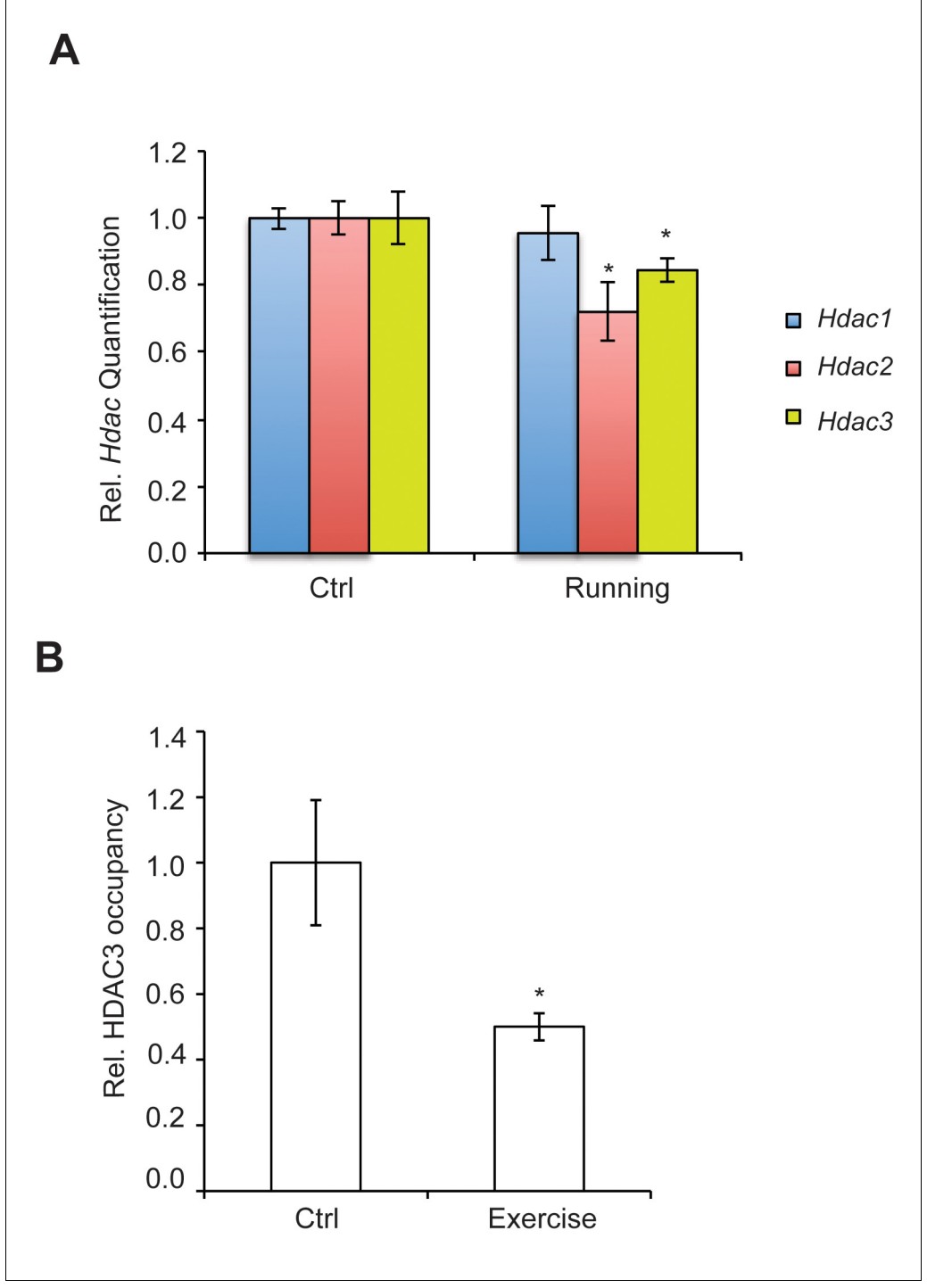

**Figure 3.** Exercise affects *Hdac* expression and binding to the hippocampal BDNF promoter. (**A**) Exercise induces significant decreases in both *Hdac2* and *Hdac3* expression in the hippocampus without affecting *Hdac1* expression as measured by real-time RTPCR. The expression was analyzed from the hippocampi of 4 different control and exercise animals. Unpaired t-tests were used to measure statistical significance *p<0.05. (**B**) Exercise induces decreases in HDAC3 binding to the *Bdnf* promoter as measured by chromatin immunoprecipitation followed by real-time RTPCR. The experiment was conducted by using tissues from 7 different control and exercise animals. Unpaired t-tests were used to measure statistical significance *p<0.05.

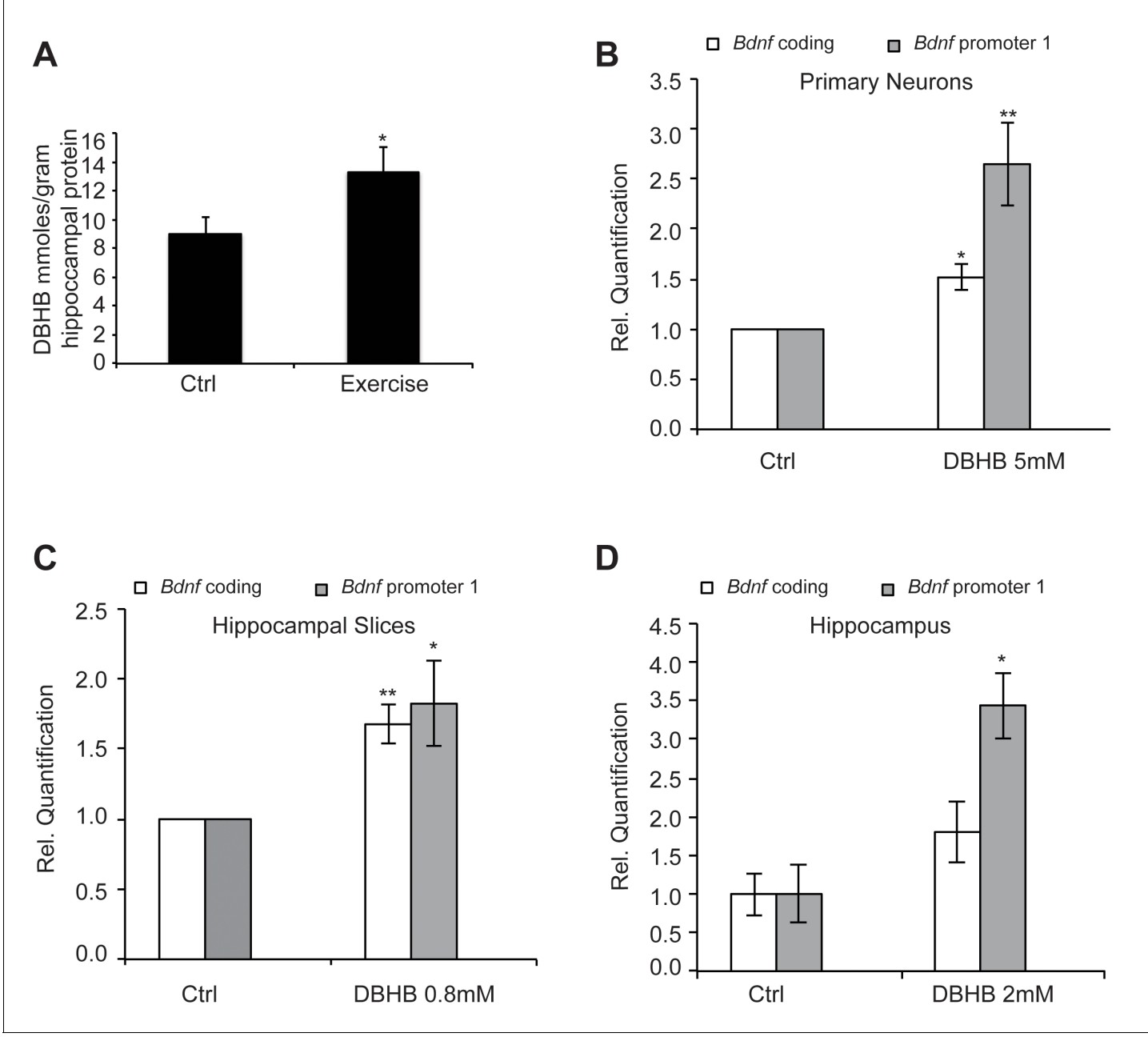

**Figure 4.** Exercise increases DBHB levels in the hippocampus. DBHB in turn can induce *Bdnf* expression in vitro in neurons and in vivo in the hippocampus (A) Exercise induced DBHB levels in the hippocampus. The number for controls and exercise hippocampi is 10 and 11 respectively. Statistical significance was analyzed by the unpaired t-test *p<0.05. DBHB amounts are expressed as millimole per gram of total hippocampal protein (B) Overnight treatment of DIV6 primary cortical neurons with 5 mM of DBHB significantly induces coding and pI *Bdnf* expression levels as measured by real-time RTPCR. For both promoters, the n number for controls and DBHB are 4 and 4 respectively. Each replicate consisted of primary neurons obtained from different cultures and treated with fresh dilutions of the compounds. Significance was measured by unpaired t-test *p<0.05 and **p<0.01. (C) Three hour treatment of hippocampal slices with DBHB (0.8 mM) induces coding and pI *Bdnf* levels as measured by real-time RTPCR. For both promoters, the n number for controls and DBHB are 5 and 5 respectively. Each replicate consisted of slices obtained from different animals and treated with fresh dilutions of the compounds. Significance was measured by unpaired t-test *p<0.05 and **p<0.01. (D) Intraventricular delivery of DBHB (2 mM) significantly induces coding and pI *Bdnf* levels as measured by real-time RTPCR. The n number for controls and DBHB are 3 and 5 respectively. Each replicate consisted of hippocampi from different animals that were subjected to the surgical procedure. Significance was measured by unpaired t-test *p<0.05.

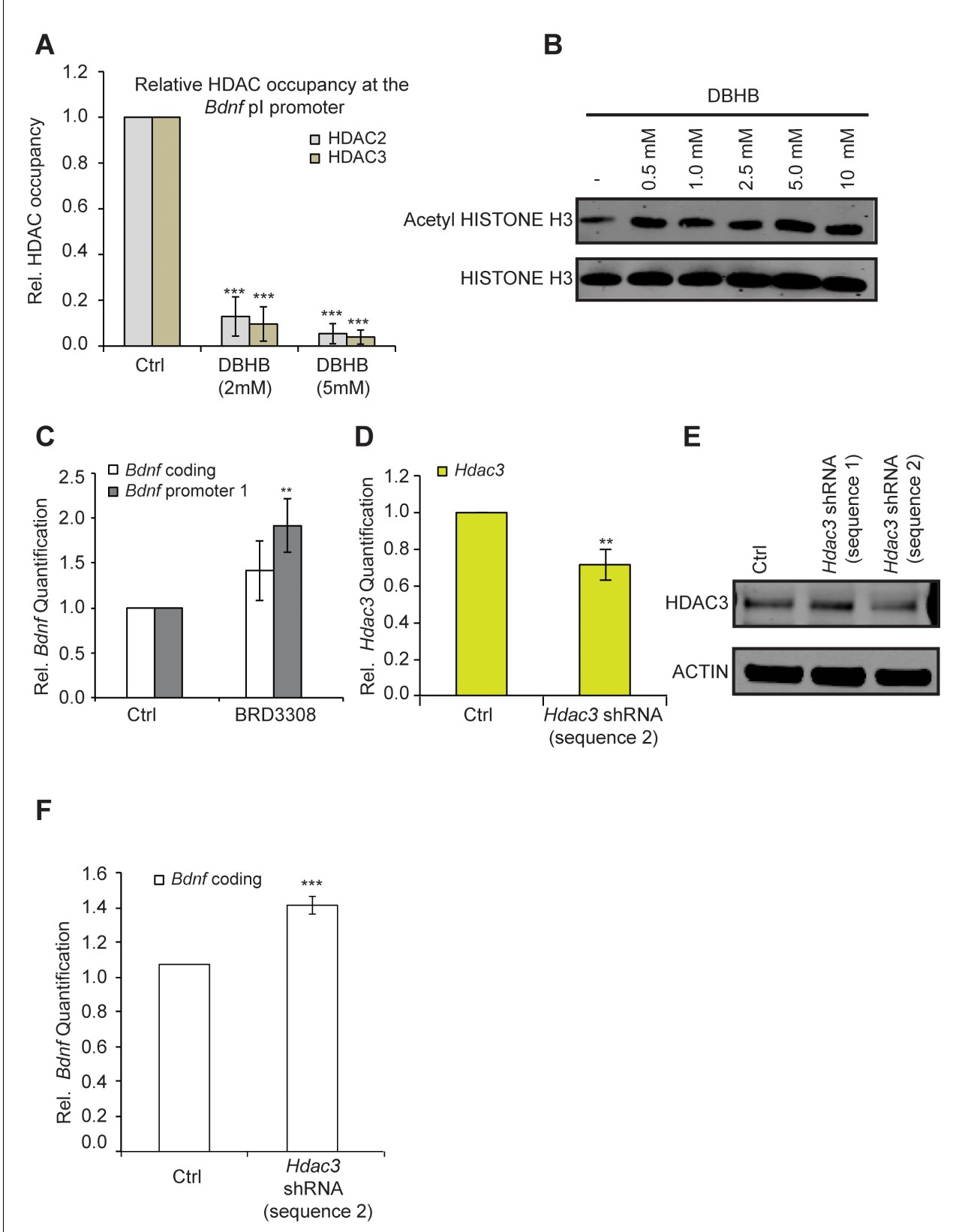

**Figure 5.** DBHB induces *Bdnf* expression by inhibiting HDAC2 and HDAC3. (**A**) DBHB (2 mM or 5 mM) treatment decreased the binding of both HDAC2 and HDAC3 on the pI promoter of *Bdnf* as measured by chromatin immunoprecipitation followed by real-time RTPCR. The number of
*Figure 5 continued on next page*

*Figure 5 continued*

chromatin immunoprecipitations for each HDAC for each treatment were 3 (control), 3 (DBHB 2 mM) and 2 (DBHB 5 mM). Statistical significance was measured by 1way anova ****p<0.001. (B) HISTONE H3 acetylation is increased in neurons upon treatment with different doses of DBHB. Representative blot shown in the figure. (C) The HDAC3 selective inhibitor BRD3308 significantly induces *Bdnf* pI expression as measured by real-time RTPCR.. For the coding promoter, the n number for controls and BRD3308 treatments are 6 and 3 respectively. For the pI promoter, the n number for controls and BRD3308 treatments are 5 and 4 respectively. Each replicate consisted of primary neurons obtained from different cultures and treated with fresh dilutions of the compounds. Significance was measured by unpaired t-test **p<0.01. (D) *Hdac3* knockdown is verified by real-time RTPCR. N = 5 represents independent times knockdown was achieved in different primary culture using nucleofection. Significance was measured by unpaired t-test **p<0.01. (E): HDAC3 knockdown is verified by western blotting. (F): HDAC3 knockdown significantly induces *Bdnf* coding gene expression as verified by real-time RTPCR. N = 5 and significance was measured by unpaired t-test *p<0.05. The *Hdac3* shRNA sequence 2 that significantly reduced HDAC3 protein levels induced *Bdnf* expression.

## DBHB induces BDNF expression

To evaluate if DBHB had an effect upon *Bdnf* expression, we treated cortical neurons, which represent an abundant source of activity-dependent BDNF. Overnight treatment with DBHB significantly induced the coding and pI-driven *Bdnf* transcripts (*Figure 4B*), consistent with the effects of exercise. Treatment of hippocampal slices with DBHB also gave similar results (*Figure 4C*). To test the ability of DBHB to induce *Bdnf* in an in vivo setting, we administered exogenous DBHB intraventricularly in mice. Gene expression analysis indicated an increase specifically in *Bdnf* promoter I activity in the hippocampi of mice receiving the intraventricular injection of DBHB as compared to saline injections (*Figure 4D*).

## DBHB induces BDNF expression through HDAC2/HDAC3 inhibition and Histone H3 acetylation

One possible explanation for the effects of exercise is that it induces DBHB accumulation in the hippocampus, which in turn inhibits class I HDACs such as HDAC2 and HDAC3 and leads to induction of *Bdnf* expression. To determine whether DBHB induction of *Bdnf* represents an HDAC-dependent mechanism, we tested whether DBHB treatment affected HDAC2 and HDAC3 occupancy at the *Bdnf* promoters. We find that treatment of primary neurons with DBHB led to decreased HDAC2 and HDAC3 binding to the *Bdnf* promoters (*Figure 5A*). This was consistent with an increase in levels of acetylated histone H3 after DBHB treatment (*Figure 5B*). Because DBHB is a potent inhibitor of HDAC3 (*Shimazu et al., 2013*) and HDAC2 has been previously shown to repress BDNF gene expression (*Guan et al., 2009*), we focused our attention on HDAC3. Inactivation of HDAC3 either by treating cortical neurons with a HDAC3 selective HDAC inhibitor BRD3308 (*Barton et al., 2014*) *Wagner et al., 2016* (*Figure 5C*) or through knockdown using shRNA targeting HDAC3 (*Figure 5D, E,& F,E*) both led to increases in *Bdnf* expression. Taken together, these results suggest that inhibition of HDAC2 and HDAC3 in the hippocampus induces *Bdnf* expression.

## DBHB links exercise induced metabolic changes in the liver to changes in gene expression in the hippocampus

While brain cells catabolize glucose for energy under normal physiological conditions, ketone bodies are utilized when blood glucose levels decrease, as observed during exercise, fasting or caloric restriction. In order to mimic the increases in DBHB in the brain observed after exercise, we injected mice with 2-deoxy-d-glucose (2-DG), a structural analog of glucose that inhibits glycolysis and increases the brain's capacity to utilize ketone bodies as fuel. 2-DG is transported by glucose transporters into the cell where it binds to, but cannot be phosphorylated by hexokinase, resulting in the inhibition of the first step of glycolysis (*Ralser et al., 2008*). This inhibition leads to the activation of a compensatory mechanism resulting in the production of ketone bodies by the liver. By activating this alternative energy pathway, 2-DG treatment induces DBHB levels in the brain. Interestingly, we found that 2-DG injection, like exercise, also induces both DBHB levels and *Bdnf* expression in the hippocampus to a similar extent (*Figure 6A and B*). Interestingly, co treatments of a DBHB transporter inhibitor (AR-C155858) with 2-DG attenuated the 2-DG-induced DBHB levels and *Bdnf* expression. Hence, these findings are consistent with a model in which exercise induces DBHB

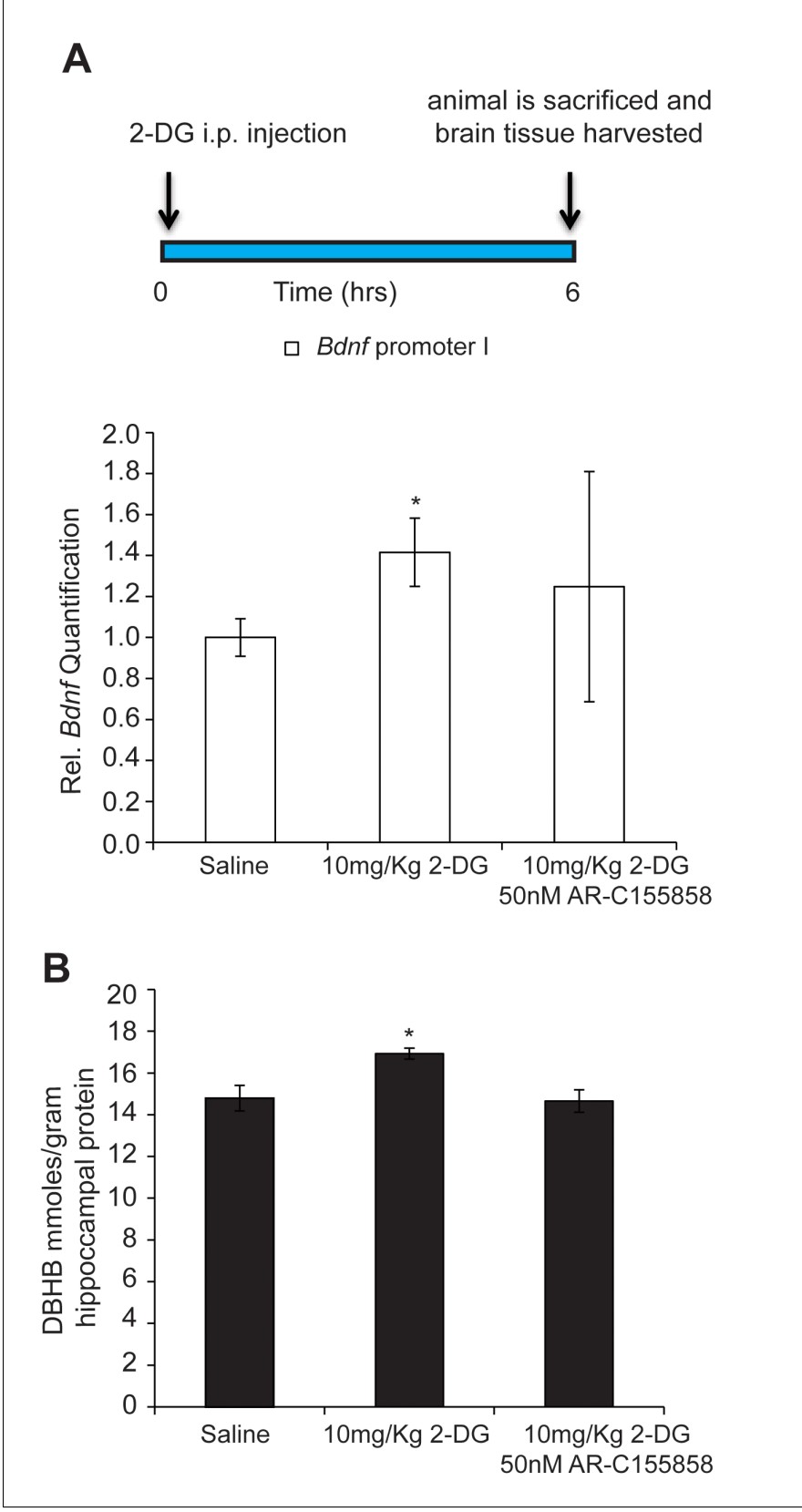

**Figure 6.** DBHB can serve as an exercise factor linking metabolic changes in response to exercise to changes in gene expression in the brain. (**A**) 2-DG treatment known to induce ketone bodies in the brain induces *Bdnf* pI
*Figure 6 continued on next page*

*Figure 6 continued*

expression in the hippocampus as measured by real-time RTPCR. This effect was blocked by a DBHB transporter inhibitor. N = 10 (Ctrl), N = 9 (2-DG injected) and N = 5 (2-DG and AR-C155858). Significance was measured by unpaired t-test *p<0.05. (B) 2-DG treatment induced DBHB levels in the hippocampus and this effect is blocked by a DBHB transporter inhibitor.

accumulation in the hippocampus (*Figure 6A and B*). DBHB stimulates histone acetylation at the *Bdnf* promoters through reduced HDAC2 and HDAC3 occupancy. This results in an increase in *Bdnf* gene transcription which reflects an epigenetic mechanism after exercise.

## DBHB induces neurotransmitter release in the hippocampus

To investigate the consequences of an increase of DBHB in the hippocampus, we conducted electrophysiological measurements in hippocampal slices to evaluate DBHB's effects on synaptic transmission. We incubated hippocampal slices with DBHB (0.8 mM) and then carried out field recording of post-synaptic potentials (fEPSPs) in CA1 evoked by stimulating the Schaffer collateral fibers (*Yano et al., 2006*). Paired pulsed facilitation (PPF) was measured as the ratio of fEPSP slopes in response to two stimuli delivered to the Schaffer collateral inputs. PPF was followed after interstimulus intervals of 20–40 milliseconds. Incubation of hippocampal slices with DBHB (0.8 mM, 3 hrs) increased the fEPSP slope which was blocked by K252a (200 nM). Consistent with these results, DBHB decreased paired pulse ratio in a K252a-dependent manner suggesting a pre-synaptic modulation mediated by TrKB signaling (*Figure 7A and B*). The actions of BDNF upon synaptic transmission are known to result in an increase in the frequency of synaptic currents, indicating a presynaptic role of BDNF (*Park and Poo, 2013*). In fact, BDNF rapidly increases spontaneous neurotransmitter release in hippocampal neurons (*Jovanovic et al., 2000*; *Li et al., 1998b*), which require TrkB receptors localized at pre-synaptic sites (*Li et al., 1998*). These measurements indicated that DBHB treatment enhanced glutamatergic transmission at CA3-CA1 synapses (*Figure 7*) and that this DBHB-induced effect was blocked by K252a, a frequently used inhibitor of TrkB receptors in synaptic transmission experiments (*Chen et al., 2015*; *Li et al., 1998b*; *Takei et al., 1998*). K-252a at 200 mM concentration specifically inhibits Trk receptors and not other tyrosine kinase receptors (*Berg et al., 1992*). Hence the presynaptic enhancement by DBHB is dependent upon the BDNF TrkB receptors. These findings are consistent with an increase of BDNF by DBHB in the hippocampus and furthermore indicate there are additional physiological outcomes mediated by DBHB that involve an increase in neurotransmitter release.

## Discussion

These results provide a link between running exercise, the ketone body DBHB and *Bdnf* gene expression. Previous work with DBHB showed it was an effective neuroprotective agent in Huntington's disease (*Lim et al., 2011*) and Parkinson's disease (*Kashiwaya et al., 2000*; *Tieu et al., 2003*), affecting striatal and dopaminergic neurons, respectively. It is highly conceivable that DBHB might act to increase the levels of BDNF, which can rescue neurons that are vulnerable in Huntington's and Parkinson's disorders (*Autry and Monteggia, 2012*). Indeed, treatment of Alzheimer's disease 3xTgAD mice with 2-DG to induce ketone bodies delayed the progression of bioenergetic deficits in the brain and the associated β-amyloid burden. Similar to our experiments, 2-DG was capable of inducing BDNF in 3xTgAD mice (*Yao et al., 2011*). Brain is a major source of the secretion of BDNF after exercise (*Rasmussen et al., 2009*). Moreover, changes in the levels of BNDF have been suggested in the pathophysiology of schizophrenia, addiction and eating disorders (*Autry and Monteggia, 2012*). A number of studies have shown that exercise can also improve depressive-like behavior through increased levels of hippocampal BDNF, which can enhance plasticity and synaptogenesis and reduce neurodegeneration (*Aguiar et al., 2011*; *Cotman et al., 2007*; *Duman and Monteggia, 2006*; *Lu et al., 2013*; *Marais et al., 2009*; *Martinowich et al., 2007*; *Russo-Neustadt et al., 2000*). Given that histone acetylation in the hippocampus and cortex is associated with effects on

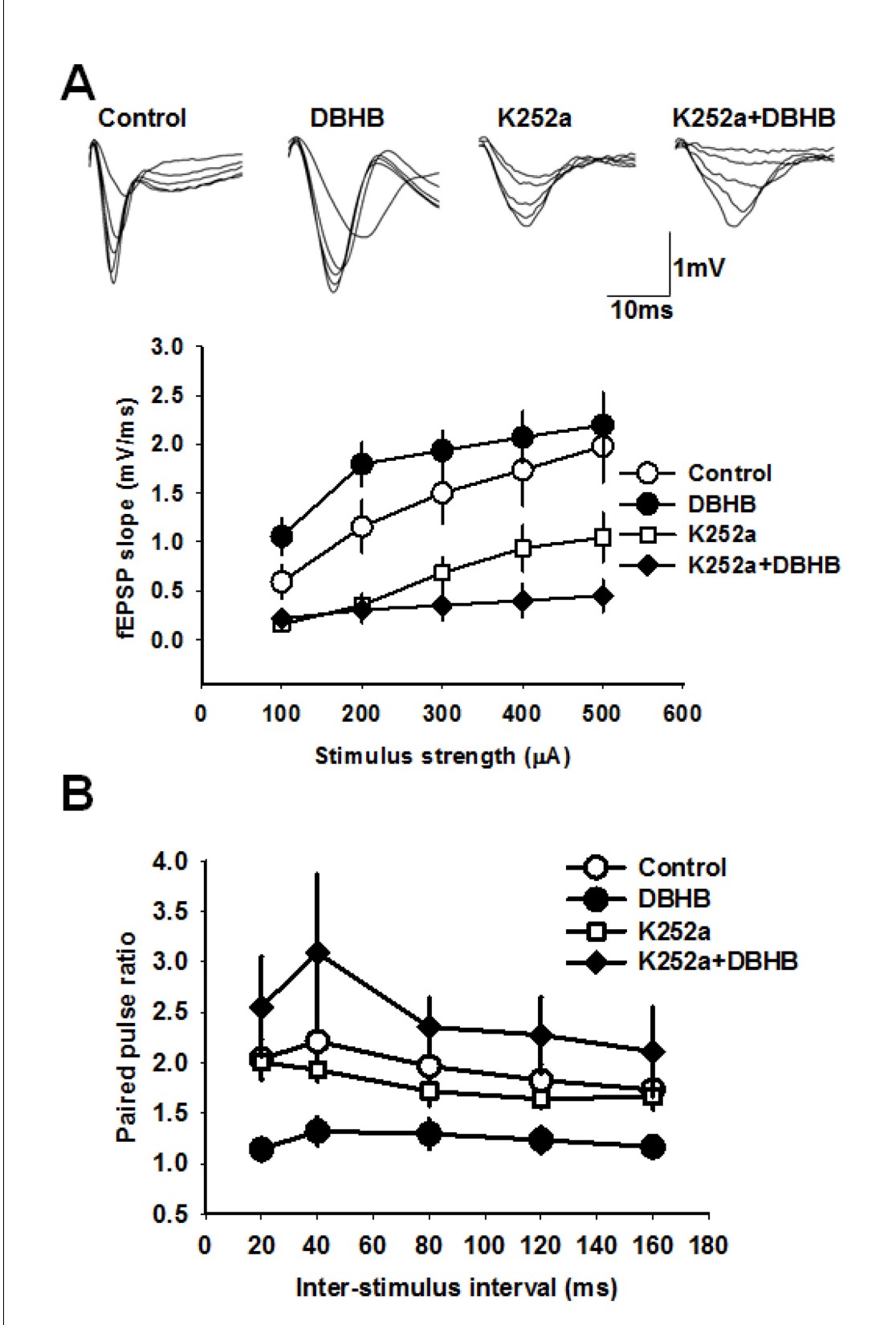

**Figure 7.** DBHB increases glutamatergic transmission at the CA3-CA1 synapses in a TrkB-sensitive manner. (**A**) Average fEPSP slope in control (6 slices/ 6 mice), DBHB (6 slices/6 mice), K252a (6 slices/3 mice), and K252a+DBHB (6 slices/3 mice) groups. Two-way repeated measures ANOVA showed
*Figure 7 continued on next page*

*Figure 7 continued*

significant difference between groups ($F_{3,20}$ = 10.13, p<0.001). Upper panel shows examples fEPSP traces. (**B**) Average paired pulse ratio in control (6 slices/6 mice), DBHB (6 slices/6 mice), K252a (6 slices/3 mice), and K252a+DBHB (6 slices/3 mice) groups. Two-way repeated measures ANOVA showed significant difference between groups ($F_{3,20}$ = 4.03, p = 0.022).

learning and memory, the ketone body DBHB serves as a metabolic signal to link environment changes to epigenetic effects on the transcription of neurotrophic factors, such as BDNF.

Ketone bodies are widely distributed from the liver to the heart, muscle and the brain after fasting, dieting and intense exercise. When glucose levels are reduced, ketone bodies, produced in the liver from fatty acids in the form of DBHB and acetoacetate, serve as an energy source. In the brain, the levels of ketone bodies can reach very high levels (1–5 mM) (*Mitchell et al., 1995*; *Robinson and Williamson, 1980*). A number of studies have demonstrated neuroprotective effects of ketone bodies in neurodegenerative and neuronal activity, such as under epileptic conditions (*Maalouf et al., 2009*).

It is probable there are multiple mechanisms that are involved in the effects of exercise in increasing trophic factor expression. For example, it has been reported that exercise induces *Bdnf* expression through the induction of hippocampal expression of Fndc5 (*Wrann et al., 2013*), a PGC-1α and ERRα -dependent myokine. Once BDNF protein levels increase, TrkB signaling in turn inhibits *Fndc5* expression in a negative feedback mechanism. We found prolonged exercise decreased *Fndc5* expression levels consistent with elevated BDNF protein in the hippocampus (*Wrann et al., 2013*). However, DBHB did not induce *Pgc-1α, Fndc5*, or *Erra* mRNA in primary neurons (data not shown), suggesting that there are alternative ways of affecting BDNF levels. The complexity of the BDNF gene and its alternative promoters and splicing events (*Haberland et al., 2009*) indicates that several regulatory mechanisms must exist to explain its many functions in synaptic transmission and neuronal survival (*Park and Poo, 2013*). Interestingly, exercise induces *Bdnf* expression in the hippocampus and not in all the other regions of the brain. How this specificity is achieved is not clear. One possible explanation suggests that exercise may regulate the levels or functions of ketone body transporters in different regions of the brain. Indeed, acute exercise induces *Mct2* transporter expression in the hippocampus immediately after and up to 10 hr post-exercise. This exercise-induced upregulation of *Mct2* was associated with increases in BDNF and TrkB levels (*Takimoto and Hamada, 2014*).

Our finding of an endogenous metabolite, DBHB, which upregulates *Bdnf* transcription in the hippocampus, provides one explanation linking the effects of exercise and peripheral metabolism to changes in epigenetic control and gene expression in the brain. There are multiple links between metabolic changes resulting from exercise and epigenetic modulation. For example, it has been reported that the blood levels of some Kreb's cycle intermediates such as α-ketoglutarate are increased after exercise (*Leibowitz et al., 2012*). Interestingly, α-ketoglutarate is an essential co-substrate for jumanji histone demethylases as well as for the tet proteins (DNA demethylases). Another molecule that may play a critical role in linking metabolic changes induced by exercise to epigenetic modulation is acetyl CoA which serves as the source of the acetyl group for all acetylation reactions catalyzed in the body including those catalyzed by histone acetyltransferases. Finally, coenzymes such as NAD+, or FAD produced during metabolic reactions are also required by sirtuins and the histone demethylase LSD1, respectively.

In this paper, we provide evidence that an endogenous molecule, DBHB, that crosses the blood brain barrier, is increased by physical exercise to enhance the expression of a fundamental trophic factor in the brain and in turn affect synaptic transmission (*Figure 8*). Further studies aiming at identifying molecules that can also serve the dual purpose of an energy fuel and epigenetic modulator will help us accumulate additional members of the "exercise pill". The identification of these molecules is of great interest as many people afflicted with depression or with neurodegenerative diseases are likely to benefit from the ability of exercise to stimulate BDNF through small metabolites, such as DBHB. The involvement of ketone bodies in many other syndromes, such as glucose utilization, diabetes and epilepsy, suggests they represent vital molecules with broad metabolic effects upon chromatin and gene expression.

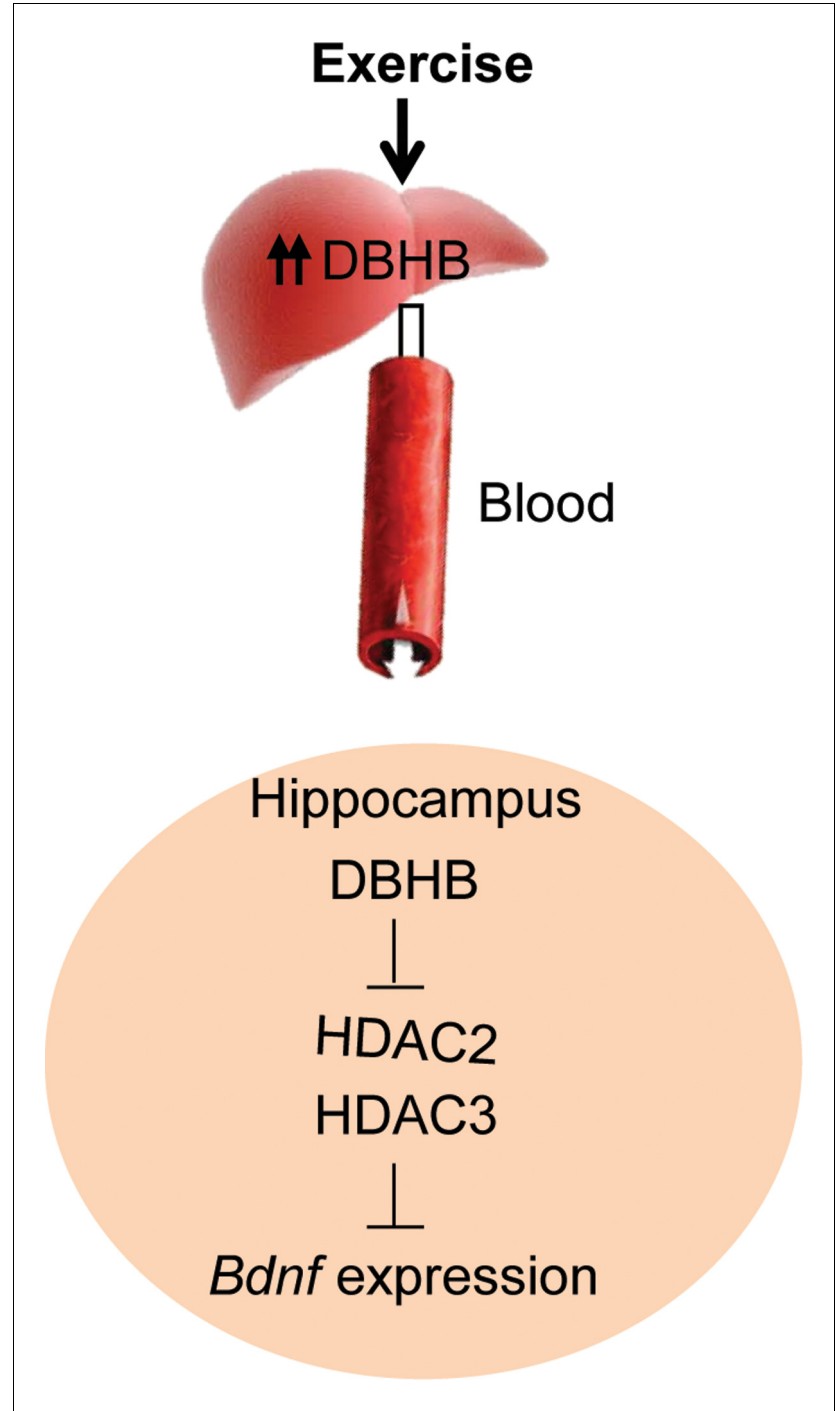

**Figure 8.** A proposed model by which exercise induces *Bdnf* expression in the hippocampus. Exercise induces DBHB synthesis in the liver. DBHB is transported through the circulation to peripheral organ including the brain. In the hippocampus, DBHB induces *Bdnf* expression through a mechanism involving HDAC inhibition. This induction in turn mediates exercise's positive effects on memory, cognition and synaptic transmission.

# Materials and methods

### Exercise paradigm

Male mice were individually housed with food and water ad libitum, lights on: 6 AM and lights off 6 PM. They were divided into 2 groups (Control and Running, n = 10 each); Control mice and running animals were housed with free access to a running wheel. Running distance and time were monitored (Coulborn Instruments). Animals were sacrificed after 30 days and hippocampus and cortex were collected and immediately frozen on dry ice. Animal care and use was in accordance with the guidelines set by the National Institutes of Health and the New York State Department of Health

### Cell culture

Immature primary cortical neurons were obtained from C57BL/6 mice [embryonic day 17 (E17)] as previously described (*Ratan et al., 1994*). Mature cortical neurons were maintained in Neurobasal media (Invitrogen) supplemented with MACS NeuroBrew- 21 (Miltenyi Biotec), and Glutamax (Invitrogen).

### Cell treatment

Primary neurons wereisolated as described above and 0.5 million cells were plated in six well plates. On Day 6, cells were treated with different concentrations of DBHB overnight or HDAC inhibitors for 5 hr. DBHB was prepared as 1 M stock in PBS and used at a final concentration of 5 mM. suberoylanilide hydroxamic acid (SAHA), Acetyldinaline (CI-994), Entinostat (MS-275), BRD3308 and BRD4097 were prepared as 10 mM stocks in DMSO and used at a final concentration of 10 µM. Trichostatin A (TSA) was prepared as a 670 µM stock and used at a final concentration of 0.67 µM.

### RNA extraction and Real Time PCR

Total RNA was prepared from primary cortical neurons or hippocampi usingthe NucleoSpin RNA II Kit (Clontech) according to the manufacturer's protocol. Real-time PCRs wereperformed using standard PCR protocol. Details of the Primers used are provided below:

### Primer sequence (5'-3'):

*Bdnf* coding (Fwd): GCGGCAGATAAAAAGACTGC
  *Bdnf* coding (Rev): GCAGCCTTCCTTGGTGTAAC
  *Bdnf* Var (Rev): GCCTTCATGCAACCGAAGTA
  *Bdnf* Var I (Fwd): CAGGACAGCAAAGCCACAAT
  *Gapdh* (Fwd): CTCTCTGCTCCTCCCTGTTC
  *Gapdh* (REV): CCGACCTTCACCATTTTGTC
  *Hdac3* (Fwd): ATGCCTTCAACGTGGGTGAT
  *Hdac3* (Rev): CCTGTGTAACGGGAGCAGAAC

### DBHB measurement

DBHB levels were measured using a DBHB Assay kit (MAK041, Sigma) according to manufacturer's protocol.

### Immunoblot analysis

*To determine HDAC3, acetyl HISTONE H3, and HISTONE H3 protein levels, total cell proteins were prepared by lysing cells in RIPA-B (1% Triton X-100, 1% SDS, 50 mM Tris-Cl, pH 7.4, 500 mM NaCl and 1 mM EDTA) in the presence of protease inhibitors (Sigma), the proteasome inhibitor MG-132 (Sigma) and phosphatase inhibitors (Sigma) followed by benzonase nuclease (Sigma) digestion for 15 min. Samples were boiled in Laemmlibuffer and electrophoresed under reducing conditions on NuPAGE Novex 4–12% for Bis-Tris Gel polyacrylamidegels (Invitrogen). Proteins were transferred to a nitrocellulose membrane(Bio-Rad) by electroblotting. Nonspecific binding was inhibitedby incubation in Odyssey blocking buffer (LI-COR Biosciences). Antibodies against HISTONE H3 (Cell signaling) and acetylated HISTONE H3 (Millipore), HDAC3 (ab16047, Abcam) and β-ACTIN (AC-74; Sigma-Aldrich) were diluted 1:1000, 1:1000, 1:1000; and 1:10,000, respectively, in odyssey blocking buffer and the membranes were incubated overnight at 4°C. Fluorophore-conjugated Odyssey*

IRDye-680 or IRDye-800 secondary antibody (LI-COR Biosciences) was used at 1:10,000 dilution followed by incubation for 1 hr at room temperature. Finally, proteins were detected using an Odyssey infrared imaging system (LI-COR Biosciences). To determine BDNF protein levels, total hippocampal proteins were *prepared by homogenizing tissues in* RIPA-B in the presence of protease inhibitors, the proteasome inhibitor MG-132 and phosphatase inhibitors followed by benzonase nuclease digestion for 15 min. Samples (100μg) were boiled in Laemmlibuffer and electrophoresed under reducing conditions on NuPAGE® Novex 12% Bis-Tris Gel polyacrylamidegels. Proteins were transferred to a nitrocellulose membrane(Bio-Rad) by electroblotting. The membranes were washed twice with 1X PBS followed by 30 min incubation with 2.5% glutaraldehyde/PBS. The membranes were washed twice again with PBS and then with TBS-T. Nonspecific binding was inhibitedby incubation in 5% milk/TBS-T followed by incubation with the BDNF antibody (N-20, Santa Cruz) for 2 hr at room temperature. A peroxidase conjugated goat anti-rabbit or anti mouse was used for BDNF or actin detection, respectively. The membranes were detected by chemiluminescence.

## Hippocampal slices

Animal care and use was in accordance with the guidelines set by the National Institutes of Health and the New York State Department of Health. 5-week old male mice were anesthetized and decapitated. The brain was dissected and placed in 4°C cutting buffer (126 mM sucrose, 5mM KCl, 2 $CaCl_2$, 2 mM $MgSO_4$, 26 mM $NaHCO_3$, 1.25 mM $NaH_2PO_4$, and 10 mM D-glucose, pH 7.4). The hippocampus was dissected and submerged in ice-cold cutting buffer and cut horizontally into 300-μm sections, which were immediately placed in recovery solution {50% cutting solution/ 50% artificial cerebrospinal fluid (ACSF)} and oxygenated (95% $CO_2$-5% $O_2$) for 20 min. The slices were then transferred in to ACSF oxygenated chambers and treated with DBHB (0.8 mM concentration) for 3 hr after which RNA was extracted for Real-Time RT PCR analysis.

## Intraventricular delivery of *DBHB*

All surgeries were performed in accordance to IACUC rules. Mice were anesthetized under isoflurane and placed on a stereotaxic frame (David Kopf instruments, CA). The body temperature of the animals was maintained at 37°C using a homeothermic blanket. DBHB (2 mM and 5 mM; 5 μl) was delivered by a Hamilton syringe at a flow rate of 0.5 μl/min using a nanomite syringe pump (Harvard apparatus, MA). The stereotaxic coordinates relative to bregma were as follows: AP, -0.46 mm; L, -1.20 mm; DV, 2.20 mm. (Paxinos and Watson, 1998). In sham control animals, 5 μl of saline (vehicle) were infused. Proper post-operative care was taken until the animals recovered completely. Mice were sacrificed after 6 hr following the injection and the different brain regions were dissected.

## Hdac3 short hairpin RNA knockdown

*Hdac3* (NM_010411)

short hairpin RNA (shRNA) clone (TRCN0000039391, 5' CCGGGTGTTGAATATGTCAAGAGTTC TCGAGAACTCTTGACATATTCAACACTTTTTG 3'; Sigma) and Non-Target shRNA Control Vector (Sigma) were introduced into immature primary cortical neurons (E17) using the Amaxa mouse Neuron Nucleofector kit as directed by the manufacturer (Lonza). On Day 6, HDAC3 knockdown was confirmed by Real-time RTPCR and whole-cell lysate Western blots.

## Chromatin immunoprecipitation

The Ez-Magna ChIP assay kit was used as directed by the manufacturer (Millipore). Briefly, primary cortical cells were crosslinked with 1% formaldehyde at 37°C for 7 min. Cells were then sonicated using the Bioruptor (Diagenode) and immunoprecipitated with primary antibodies (10 μg). The crosslinking was reversed, and the DNA was isolated on the columns provided by the kit. Shearing size was determined to be between 150 and 1000 bp. Real-time PCR was conducted with primers targeted to the BDNF pI promoter (TGATCATCACTCACGACCACG and CAGCCTCTCTGAGCCAG TTACG) and SYBR Green PCR Master mix (Applied Biosystems). Each experiment was conducted at least three times by crosslinking cells from different primary cortical neuron preparations.

## 2- deoxy-d-glucose injections

mice received intraperitoneal injections of 10mg/Kg of 2-DG or saline. They were left to recover for 6 hrs after which the animals were sacrificed and the tissue was harvested.

## Electrophysiology

2–4 months old C57BL6 male mice were anesthetized by pentobarbital anesthesia. After decapitation, hippocampi were isolated in ice-cold artificial cerebrospinal fluid (ACSF) containing the following (in mM), NaCl (118), KCl (4.5), glucose (10), $NaH_2PO_4$ (1), $CaCl_2$ (2) and $MgCl_2$ (2) (aerated by 95%O2/5% CO2, pH adjusted to 7.4 with $NaHCO_3$). Hippocampal slices (300 μm) were prepared on a vibratome (Campden Instruments) and maintained at room temperature for 1 hr in a brain slice keeper. Brain slices were incubated in vehicle (0.00007% DMSO), K252a (200 nM), DBHB (0.8 mM) or combination of K252a and DBHB for 3 hrs at room temperature before recording CA1 field excitatory post-synaptic potentials (fEPSPs) in a recording chamber perfused with ACSF at 32℃. fEPSPs were evoked by stimulation of the Schaffer collateral fibers using a concentric bipolar electrode (FHC). Input-output data were generated by plotting fEPSP slope in response to 100, 200, 300, 400 and 500 μA stimulation. Paired pulse ratio was measured as ratio of second fEPSP slope to first EPSP slope at 20, 40, 80, 120 and 160 ms inter-stimulus interval. A response approximately 35% of the maximum evoked response was used for studying paired pulse ratio. Data were acquired using pCLAMP 10 program and MultiClamp 700B amplifier (Molecular Devices). Data analysis was carried out using Clampfit program.

## Statistical analysis

unpaired t-test, 1way or 2way ANOVA followed by the Dunnett or Bonferroni post tests respectively were used to measure statistical significance. $p < 0.05$ was considered to be statistically significant.

## Acknowledgements

We appreciate the generosity of Henriette van Praag and Tonis Timmusk in providing materials for this study. We also appreciate experimental suggestions from Amit Kumar and technical assistance of Jean Karam. Support was provided by the NIH (NS21072, AG025970) to MVC and (HD076914) to IN. SFS is supported by grants from the Lebanese American University School of Arts and Sciences and The National Council for Scientific Research (CNRS) of Lebanon (grant #699). Support for this work was also provided by Ofer Nemirovsky. MVC and SFS conceived the study and wrote the manuscript. SFS performed the experiments with the help of JKH, RAH, LEH, EAH, TS, DU and SSK. EBB and RRR provided reagents and advice about the specificity of HDAC inhibition and IN carried out the electrophysiological analyses of hippocampal slices.

## Additional information

### Competing interests

MVC: Reviewing editor, *eLife*. EBH: is a consultant for KDAc Therapeutics which has licensed compounds from the Broad Institute. The other authors declare that no competing interests exist.

### Funding

| Funder | Grant reference number | Author |
|---|---|---|
| National Council for Scientific Research | #699 | Sama F Sleiman |
| Lebanese American University | SRDC seed grant | Sama F Sleiman |
| National Institutes of Health | HD076914 | Ipe Ninan |
| National Institutes of Health | NS21072 | Moses V Chao |
| National Institutes of Health | AG025970 | Moses V Chao |

The funders had no role in study design, data collection and interpretation, or the decision to submit the work for publication.

## Author contributions

SFS, Conception and design, Acquisition of data, Analysis and interpretation of data, Drafting or revising the article, Contributed unpublished essential data or reagents; JH, RA-H, LEH, EAH, DU, SSK, Acquisition of data, Analysis and interpretation of data; TS, Acquisition of data; EBH, Analysis and interpretation of data, Contributed unpublished essential data or reagents; RRR, Analysis and interpretation of data; IN, Acquisition of data, Analysis and interpretation of data, Drafting or revising the article; MVC, Conception and design, Analysis and interpretation of data, Drafting or revising the article

## Author ORCIDs

Moses V Chao, http://orcid.org/0000-0002-6969-3744

## Ethics

Animal experimentation: This study was performed in strict accordance with the recommendations in the Guide for the Care and Use of Laboratory Animals of the National Institutes of Health and the New York State Department of Health. All of the animals were handled according to approved institutional animal care and use committee (IACUC) protocols of New York University (Approved Protocol (#140601) All surgery was performed under sodium pentobarbital anesthesia, and every effort was made to minimize suffering.

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
