## [Decision Letter]

Thank you for submitting your article "Exercise promotes the expression of BDNF through the action of the ketone body Β-hydroxybutyrate" for consideration by *eLife*. Your article has been favorably evaluated by K VijayRaghavan (Senior editor) and three reviewers, one of whom is a member of our Board of Reviewing Editors.

The reviewers have discussed the reviews with one another and the Reviewing Editor has drafted this decision to help you prepare a revised submission.

Summary:

Sleiman and colleagues describe results from a series of studies linking rises of ketone bodies during exercise and increased BDNF expression in the hippocampus. In particular, they demonstrate that β-hydroxybutyrate (BHB) rise in their exercise paradigm and this leads to increased BDNF. They mimic this rise in vitro by treating hippocampal slices and report an increase in BDNF. Finally, they used ICV injections of BHB to demonstrate that this also could activate BDNF transcription. The proposed mechanism suggests an exercise-induced elevation in circulating levels of BHB inhibits the repressing effects of HDAC2 and 3 on the Bdnf promoter 1. Collectively, the studies are well described and provide interesting results. Indeed, the results will likely be of wide interest in the fields of neuroscience, exercise physiology and metabolic research.

Essential revisions:

Because exercise can induce BDNF expression through BHB-independent mechanisms, the physiological relevance of this pathway is not clear. Studies informing the necessity of this pathway on the beneficial effects of hippocampal BDNF on cognitive function, antidepressant efficacy or synaptic plasticity would strengthen the conclusions of the manuscript.

The Western data presented in Figure 1 needs to be quantified before reaching the conclusion that BDNF levels are elevated, especially as the samples from exercise animals appear to be over loaded.

The authors made the initial observation that exercise induces BDNF expression driven by both promoters 1 and 2. It is not clear why subsequent experiments only focus on promoter 1.

Figure 1 convincingly shows that exercise increases hippocampal expression of mRNA for the promoter I BDNF transcript, but the data seemingly indicates that there is no significant effect on the levels of expression detected with primers within the translated region of the transcript, presumably because promoter I does not account for a major fraction of BDNF expression. I think the authors need to discuss this feature of the data. Concern about the issue is heightened by the western blot data in Figure 1, where the upregulation of BDNF protein levels is not entirely convincing. (One of 3 replicates for the Exercise condition seems no higher than one of the 2 replicates for the control condition). Densitometric quantification, with normalization against the actin band, might dispel this concern.

Some of the data in Figure 2 and Figure 3 are compelling. However, the data in 3B, interpreted to indicate that exercise decreases the HDAC3 occupancy of BDNF promoter I, does not provide statistical analysis, and the statistical significance is unclear.

The evidence (Figure 4) that exercise increases BHB in brain seems quite convincing. However, in the Discussion, the authors cite published evidence that brain BHB levels may reach 1-5 mM, yet they report levels of 10-15 mM (calculated assuming a brain density of 1.0 g/ml), and I found two studies reporting ketogenic diet or alcoholic ketoacidosis producing maximum brain β-hydroxybutyrate levels of 180 and 120 μM (Samala et al., 2011 and Schrieber and Ungar, 1984). The dose of ICV injections of the BHB needs to be justified. Is it known whether these doses affect metabolic function in key peripheral tissues? Does this dose affect other brain regions?

---

## [Author Response]

Essential revisions:

Because exercise can induce BDNF expression through BHB-independent mechanisms, the physiological relevance of this pathway is not clear. Studies informing the necessity of this pathway on the beneficial effects of hippocampal BDNF on cognitive function, antidepressant efficacy or synaptic plasticity would strengthen the conclusions of the manuscript.

We thank the reviewers for this pertinent comment. In order to address this comment, we have performed two different types of experiments to address the physiological relevance of DBHB.

First, we injected 2- Deoxy- D-glucose (2-DG) alone or in combination with a DBHB transporter inhibitor, AR-C 155858 and assessed both BDNF promoter I expression as well as DBHB levels. DBHB levels increase when there is a switch from glucose to fatty acid metabolism. We have found that glycolytic inhibition by 2-DG induces both BDNF promoter π expression and DBHB levels in the hippocampus, whereas injecting 2-DG along with AR-C155858 attenuated both DBHB levels and the induction of BDNF π expression in the hippocampus (Figure 6). These results suggest that increased DBHB levels in the hippocampus by physiologically manipulating glucose metabolism can contribute to BDNF promoter 1 induction by HDAC2/3 inhibition. This is important because exercise mediates its positive effects on learning and memory through BDNF expression (Erickson et al., 2011). Our results showing that DBHB accumulation in the hippocampus can affect BDNF expression suggest a role for DBHB in mediating the learning and memory effects through BDNF expression.

In a second physiological approach, we have also found that DBHB induces presynaptic glutamate release, which is dependent upon BDNF/TrkB signalling (Figure 7). Hippocampal slices were prepared and incubated with DBHB for 3 hr before measurements of CA1 field excitatory post-synaptic potentials (EPSPs) after stimulation of Schaffer collaterals. Pair pulse facilitation (PPF) due to an accumulation of residual Ca^2+^ in the presynaptic terminal was observed and is indicative of enhanced transmitter release. This DBHB induced response could be blocked by treatment with K252a, a frequently used inhibitor of TrkB receptors in synaptic transmission experiments (Chen et al., 2015; Li et al., 1998; Takei et al., 1998). These findings are consistent with an increase of BDNF by DBHB in the hippocampus and furthermore indicate that one physiological outcome of DBHB is an increase in neurotransmitter release through a TrkB dependent mechanism.

Taken together, these results suggest that the production of DBHB is indeed involved in the induction of BDNF expression and that the induced and activated BDNF pathway plays important roles in mediating the effects of DBHB on glutamate release.

The Western data presented in Figure 1 needs to be quantified before reaching the conclusion that BDNF levels are elevated, especially as the samples from exercise animals appear to be over loaded.

We have quantified the western blot presented in Figure 1 comparing BDNF levels from the hippocampi of control and exercise mice and the quantification is presented as Figure 1. The western blots were quantified using image J software. The quantification shows a significant increase in BDNF protein levels in hippocampi of exercise animals as compared to ctrl animals. The significance was assessed by student T-test and p-value< 0.05. These results are consistent with previously published results showing that exercise induces BDNF levels in the hippocampus using immunofluorescence and Elisa techniques (Cotman and Berchtold, 2002).

The authors made the initial observation that exercise induces BDNF expression driven by both promoters 1 and 2. It is not clear why subsequent experiments only focus on promoter 1.

Figure 1 convincingly shows that exercise increases hippocampal expression of mRNA for the promoter I BDNF transcript, but the data seemingly indicates that there is no significant effect on the levels of expression detected with primers within the translated region of the transcript, presumably because promoter I does not account for a major fraction of BDNF expression. I think the authors need to discuss this feature of the data. Concern about the issue is heightened by the western blot data in Figure 1, where the upregulation of BDNF protein levels is not entirely convincing. (One of 3 replicates for the Exercise condition seems no higher than one of the 2 replicates for the control condition). Densitometric quantification, with normalization against the actin band, might dispel this concern.

We have quantified the western blots and included the quantification in Figure 1 as mentioned above. Indeed, BDNF protein levels are induced in the hippocampi of exercise animals as compared to control animals. We have found that both promoter 1 and 2 are significantly induced by exercise. Even though the coding exon was not significantly induced, we observe a trend suggesting the increase. This trend is consistent with the western blot quantification data. Indeed, DBHB also induces promoter 2 (Data not shown) as well as the coding fraction (Figure 4). We focused on promoter 1 because previous microarray data (Tong et al., 2001) have detected an increase in promoter 1 in exercise rats as compared to control rats.

Some of the data in Figure 2 and Figure 3 are compelling. However, the data in 3B, interpreted to indicate that exercise decreases the HDAC3 occupancy of BDNF promoter I, does not provide statistical analysis, and the statistical significance is unclear.

We have increased the n number from 3 to 7 for each group for theexperiment and included statistical data. The results in Figure 3 suggest that exercise significantly (p<0.05) decreases HDAC3 binding to the BDNF promoter 1.

The evidence (Figure 4) that exercise increases BHB in brain seems quite convincing. However, in the Discussion, the authors cite published evidence that brain BHB levels may reach 1-5 mM, yet they report levels of 10-15 mM (calculated assuming a brain density of 1.0 g/ml), and I found two studies reporting ketogenic diet or alcoholic ketoacidosis producing maximum brain β-hydroxybutyrate levels of 180 and 120 μM (Samala et al., 2011 and Schrieber and Ungar, 1984). The dose of ICV injections of the BHB needs to be justified. Is it known whether these doses affect metabolic function in key peripheral tissues? Does this dose affect other brain regions?

The DBHB levels are expressed in mmole/gram. The grams in the denominator refer to the normalization to the total amount of hippocampal protein and not the grams of hippocampal tissue. We have clarified this point in the graph in Figure 4. From here, the concentration extrapolations are not accurate. The concentrations obtained, if we consider the volume used for DBHB measurement per the manufacturers’ instructions, have an average of 0.9mM (Control) and 1.3mM (Exercise). Therefore, considering our measurements and the observation that exercise modulates HDAC binding to the BDNF promoter, we chose the concentrations for ICV to be close to the measured levels and also close to the concentrations reported to have effects on HDAC inhibition (Shimazu et al., 2013).